# 2,4-Thiazolidinedione in Well-Fed Lactating Dairy Goats: I. Effect on Adiposity and Milk Fat Synthesis

**DOI:** 10.3390/vetsci6020045

**Published:** 2019-05-17

**Authors:** Shana Jaaf, Fernanda Rosa, Misagh Moridi, Johan S. Osorio, Jayant Lohakare, Erminio Trevisi, Shelby Filley, Gita Cherian, Charles T. Estill, Massimo Bionaz

**Affiliations:** 1Department of Animal and Rangeland Sciences, Oregon State University, Corvallis, OR 97331, USA; shana.jaaf@oregonstate.edu (S.J.); Fernanda.Rosa@sdstate.edu (F.R.); Johan.Osorio@sdstate.edu (J.S.O.); shelby.filley@oregonstate.edu (S.F.); gita.cherian@oregonstate.edu (G.C.); Charles.Estill@oregonstate.edu (C.T.E.); 2Department of Animal Science, University of Guilan, Kilometer 5 of Rasht-Qazvin Highway, Rasht 4199613776, Iran; moridimisagh64@gmail.com; 3Department of Animal Biotechnology, Kangwon National University, KNU Ave 1, Chuncheon 200-701, Korea; Johan.Osorio@sdstate.edu; 4Department of Animal Sciences, Food and Nutrition (DIANA), Università Cattolica del Sacro Cuore, Via Emilia Parmense, 84, 29122 Piacenza PC, Italy; erminio.trevisi@unicatt.it

**Keywords:** 2,4-thiazolidinedione, goat, milk fat, metabolism

## Abstract

*Background*: In a prior experiment, treatment of goats with the putative PPARγ agonist 2,4-thiazolidinedione (2,4-TZD) did not affect milk fat or expression of milk-fat related genes. The lack of response was possibly due to deficiency of vitamin A and/or a poor body condition of the animals. In the present experiment, we tested the hypothesis that PPARγ activation affects milk fat synthesis in goats with a good body condition and receiving adequate levels of vitamin A. *Methods*: Lactating goats receiving a diet that met NRC requirements, including vitamin A, were injected with 8 mg/kg BW of 2,4-TZD (*n* = 6) or saline (*n* = 6; CTR) daily for 26 days. Blood metabolic profiling and milk yield and components were measured including fatty acid profile. Expression of genes related to glucose and lipid metabolism was measured in adipose tissue and in mammary epithelial cells (MEC). Size of adipocytes was assessed by histological analysis. *Results*: NEFA, BHBA, and fatty acids available in plasma decreased while glucose increased in 2,4-TZD vs. CTR. Size of cells and expression of insulin signaling and glucose metabolism-related genes were larger in 2,4-TZD vs. CTR in adipose tissue. In MEC, expression of *SCD1* and desaturation of stearate was lower in 2,4-TZD vs. CTR. *Conclusions*: Overall data revealed a lack of PPARγ activation by 2,4-TZD and no effect on milk fat synthesis despite a strong anti-lipolysis effect on adipose tissue.

## 1. Introduction

Butterfat is important for milk quality, especially for taste and flavor. Butterfat is also one of the most important components in the calculation of milk price received by dairy farmers in the US. Milk fat content is highly affected by the composition of the diet, especially dietary fat and fiber.

The main components of dietary fat are long-chain fatty acids (LCFAs). They are known to have a nutrigenomic effect through modulation of transcription factors, especially peroxisome proliferator-activated receptors (PPARs) [1]. Among the three known PPAR isotypes, the PPARγ in mammary tissue of ruminants has been the most studied, and preliminary evidence supported a role of it in the regulation of milk fat synthesis [2]. Several subsequent *in vitro* studies performed on goat and bovine mammary epithelial cells confirmed the original observation [3,4,5].

Several *in vitro* studies were carried out using PPARγ agonists on dairy ruminants. The treatment of dairy cows with 2,4-thiazolidinedione (2,4-TZD), a putative PPARγ agonist, during the prepartum period did not affect milk fat percentage and either did not affect or numerically decreased milk fat yield in early postpartum cows [6]. Peripartum cows treated with the synthetic PPARγ agonist pioglitazone had a significant decrease in milk fat percentage and yield compared to cows not treated [7]. In both studies, the large decrease in plasma non-esterified fatty acids (NEFA) might partly explain the decrease in milk fat synthesis detected. In both studies, the decrease in plasma NEFA compared to the control group was between 23 and 28%, while the decrease in milk fat yield was between 8 and 11%. Plasma NEFA is an important source of preformed LCFA for milk fat synthesis and a positive relation exists between plasma NEFA and milk fat concentration [8]. Results from the above *in vitro* studies did not provide support for a positive effect of PPARγ activation on milk fat yield, but there was the confounding effect of the large decrease in NEFA, especially during the early postpartum period.

In a recent study in our laboratory, we treated dairy goats in mid-late lactation with TZD. Despite not having observed a decrease in NEFA, 2,4-TZD did not increase milk fat yield but tended to prevent milk fat decrease after induction of mammary infection [9]. Furthermore, we did not observe any large effect of 2,4-TZD on expression of classical PPARγ target genes in mammary epithelial cells (MEC) or adipose tissue, indicating that 2,4-TZD is, at best, a weak PPARγ agonist. These results confirmed the lack of effects of 2,4-TZD on expression of PPARγ target genes in adipose tissue of dairy cows during the close-up period [10]; however, another study detected a change in expression of several genes in adipose tissue after treatment with 2,4-TZD in non-pregnant dry cows [11]. Lack of activation of PPARγ by 2,4-TZD is also supported by findings from a recent study carried out in sheep with induction of milk fat depression by conjugated linoleic acid. Treatment with 2,4-TZD did not alleviate the negative effect on milk fat synthesis [12]. However, the same study detected a positive effect of 2,4-TZD on the expression of *PPARG* in mammary tissue and on lipogenic genes in adipose tissue.

In this prior goat study performed in our laboratory [9], the animals were not in an optimal body condition, and the hay-based diet was not supplemented adequately with vitamins, especially vitamin A. Vitamin A plays an important role in PPARγ activation, due to the agonistic effect of its metabolite 9-*cis*-retinoic acid on retinoic-X-receptor, the essential heterodimer of PPARs. The essential role of 9-*cis*-retinoic acid on the activation of PPARs by 2,4-TZD was supported by several *in vitro* studies carried out in bovine and caprine mammary cells [2].

For the above reasons, it remained to be determined if PPARγ plays a role in controlling milk fat synthesis *in vivo*. Therefore, it was necessary to repeat the prior experiment [9] by using dairy goats with good initial body condition and a balanced ration including the levels of vitamin A. Therefore, our hypothesis was that continuous activation of PPARγ by 2,4-TZD increases milk fat synthesis in dairy goats in a good body condition and supplemented with adequate amounts of vitamin A. The objective of the present study, which is part of a larger study, is to assess if the activation of PPARγ by 2,4-TZD increases milk fat synthesis.

## 2. Materials and Methods

### 2.1. Experimental Design and Animal Management

The Institutional Animal Care and Use Committees (IACUC) of Oregon State University (#4448) approved all experimental procedures used in this study. Figure 1 depicts the whole study encompassing two objectives, one pertinent to the present manuscript and the other related to the companion manuscript [13]. For the study, we used 12 lactating Saanen goats (mean ± SD) 52.6 ± 16.2 days in milk, 69.2 ± 7.1 kg of body weight (BW), with a 2.6 ± 0.2 body condition score (BCS), with 1.7 ± 0.6 kids, and negative to milk bacterial analysis. For the experiment, goats were housed in individual pens and randomly assigned to treatment groups after blocking for body weight (BW), milk yield, and milk components. Goats were fed twice a day as described in detail in Appendix B. Dry matter and chemical composition of individual forages were determined by standard wet chemistry techniques at a commercial laboratory (Dairy One Forage Testing Laboratory, USA) (Appendix A). The ration amount fed was calculated individually for each goat using BW and milk yield [14] (Appendix A). Animals were drenched once daily to supply the level of vitamins required as described in Appendix B. See Appendix A for each component of the drench and the total amount of each vitamin provided.

### 2.2. Treatments

After the adaptation period, the goats started to receive at 10:00 a.m. daily injections via jugular vein of 8 mg/kg of BW of 2,4-thiazolidinedione (2,4-TZD; SC-216281, Santa Cruz Biotechnology, Dallas, TX, USA) in 10 mL sterile physiological saline (002479, Henry Schein, Dublin, OH) (*n* = 6; TZD) throughout the whole study (26 days). Control group (*n* = 6; CTR) received 10 mL of sterile physiological saline without 2,4-TZD. Considering a BW of 70 kg and a molecular weight of 117.13 g/mole, it was estimated that the maximum concentration in blood of 2,4-TZD after injection was approximately 190 mM, a dose known to activate PPAR *in vitro* in the presence of 10 μM of 9-*cis*-retinoic acid [2]. An intramammary infusion with *Streptococcus uberis* in the right half of the mammary gland was performed in all goats on Day 15 of 2,4-TZD treatment (see companion papers for details [13]). The 2,4-TZD is not a compound approved by the FDA for use with animals or humans, but it is the backbone molecules used for the production of rosiglitazone and other FDA-approved compounds used to treat type II diabetes in humans [15,16].

### 2.3. Measurements, Sample Collection, and Blood Metabolites

Details are available in Appendix B. Briefly, during the first 11 days of the experiment, while goats were nursed by the kids, milk yield was measured at Days 0, 2, 5, and 10 of treatment by removing the kids at 8:00 p.m. and goats were immediately milked. The goats without the kids were milked again 12h later, and we recorded milk yield and collected milk samples. After the kids were weaned (Day 11), goats were milked twice a day, and milk yield was measured at each milking (6:00 a.m. and 6:00 p.m.) until the end of the trial. For the present manuscript, only the milk from the left quarter (i.e., the quarter that did not receive the bacteria to induce infection, see [13]) during the morning milking was used to determine the amount of milk and milk composition. Milk samples were collected for analysis of somatic cells count (SCC), lactose, fat, protein, and solids non-fat (SNF) just before 2,4-TZD injection (time 0) and then at Days 2, 5, 15, 16, 17, 20, 21, and 22 of 2,4-TZD treatment. At Days 0, 2, 5, and 15 of 2,4-TZD treatment, additional samples were collected in 15 mL sterile tubes for fatty acid analysis. Details on dry matter intake (DMI), energy-corrected milk (ECM), dairy efficiency (DE), body condition score (BCS), and milk fatty acid analysis are available in Appendix B.

Blood samples were collected prior to treatment (time 0) and then on Days 7, 9, 15, 16, 17, 18, 22, and 26 for analysis of glucose, cholesterol, urea, creatinine, NEFA, triacylglycerol (TAG), and β-hydroxybutyric acid (BHBA) (see intra- and inter-assay variation in [13,17]). The analyses were performed following the procedures described previously [17,18] using a clinical auto-analyzer (ILAB 650, Instrumentation). To check for replicability, a duplicate was performed for up to 10% of the samples selected randomly. Samples that were evident outliers or had dubious data were re-analyzed. The calculation of fatty acids available in blood was performed as mM of NEFA + 3× mM of TAG. The percent variation of available fatty acids in blood was calculated as a percentage variation compared to time 0 in each goat.

### 2.4. Adipose Biopsy and Histological Analysis, and Mammary Epithelial Cells Isolation

Subcutaneous adipose tissue was collected by a biopsy as previously described [9] from alternate sides of the tail-head at around 10:00 a.m., and samples were flash-frozen in liquid nitrogen. The biopsy was performed on the day prior to starting 2,4-TZD and after 13 days of treatment. Adipose tissue was also collected after euthanasia for all 6 goats receiving 2,4-TZD and 3 random CTR goats. Euthanasia was performed using pentobarbital. MEC were isolated from 250 mL of milk using magnetic sorting on Day 4 of treatment. Details on the materials and methods used for histological analysis of the adipose tissue and MEC isolation are available in Appendix B. The KingFisher protocol file for MEC isolation is available in Appendix A, and the pipeline for CellProfiler is available in Appendix A.

### 2.5. Reverse Transcriptase Quantitative Polymerase Chain Reaction (RT-qPCR)

RNA was extracted from adipose tissue and MEC, and RT-qPCR was performed as described in detail in Appendix B. The purity of RNA (OD260/280) was 1.84 ± 0.13 for MEC and 1.90 ± 0.12 for adipose tissue (mean ± SD). The RNA integrity number (RIN) was assessed by electrophoretic analysis using 2100 Bioanalyzer Instruments (Agilent, Santa Clara, CA, USA) at the Center for Genome Research and Biocomputing, Oregon State University. The RIN values were (mean ± SD) 5.5 ± 0.25 for adipose tissue samples and 5.6 ± 1.5 for MEC samples (5.0 ± 1.3 for negative MEC and 6.3 ± 1.4 for positive MEC; *p* = 0.10). 

The target genes selected to be evaluated in both MEC and adipose tissue were PPARγ (*PPARG*), lipoprotein lipase (*LPL*), stearoyl-CoA desaturase 1 (*SCD1*), sterol regulatory element binding factor 1 (*SREBF1*), acetyl-CoA carboxylase alpha (*ACACA*), very low-density lipoprotein receptor (*VLDLR*), fatty acid translocase (*CD36*), and long-chain fatty acid transport protein 6 (*SLC27A6*). In adipose tissue, we also measured transcription of insulin receptor substrate 1 (*IRS1*), insulin receptor (*INSR*), fatty acid synthase (*FASN*), pyruvate dehydrogenase kinase 4 (*PDK4*), phosphoenolpyruvate carboxykinase 1 (*PCK1*), glycerol-3-phosphate dehydrogenase 1 (*GPD1*), and solute carrier family 2 (facilitated glucose transporter) member 4 (*SLC2A4*). For MEC, we also measured transcription of interleukin 8 (*IL8*), fatty acid binding protein 3 (*FABP3*), cytokeratin 8 (*KRT8*), nuclear factor (erythroid-derived 2)-like 2 (*NFE2L2*), nuclear respiratory factor 1 (*NRF1*), kappa-casein (*CSN3*), and lactalbumin alpha (*LALBA*). If not already designed [9], primer-pairs were designed as previously described [9]. Details of amplicon validation and selection of internal control genes are available in Appendix B. Details of primer pairs not previously published are available in Appendix A. 

### 2.6. Statistical Analysis

For each goat an arithmetical correction was performed to obtain the same average between groups at baseline as described previously [9]. Prior to statistical analysis, data were checked for outliers using PROC REG of SAS 9.4 (SAS Institute, Inc., Cary, NC, USA). Data with a studentized *t* > 3.0 were removed. Data were analyzed with the PROC GLIMMIX. Fixed effects in the model were treatment, time, and the treatment × time interaction as main effects (or treatment, cell, and their interaction for the gene expression of MEC) and goat as random effect. The best covariance structure, either the spatial power, the autoregressive (1), or the autoregressive (1) with heterogeneous variance, for each parameter was selected using the lowest Akaike’s information criterion. Statistical significance and tendencies were declared at *p* ≤ 0.05 and *p* ≤ 0.10, respectively. For FA analysis, a Proc GLM of SAS was used. Correlations between metabolic parameters and milk-related parameters were performed using PROC CORR of SAS. Syntaxes for SAS analysis are available in Appendix B.

## 3. Results

### 3.1. Animal Performance

Dry matter intake had a tendency (*p* = 0.09) to be affected by treatment × time with a numerical greater feed intake in TZD just prior to IMI and a numerically lower feed intake in TZD vs. CTR after IMI (Figure 2). Treatment did not affect milk yield, DMI, DE, or ECM (Figure 2). Similarly, TZD compared to CTR had not significant difference in body weight, body condition score (Appendix A), milk fat (Figure 2), or any of the other milk components (Table 1).

### 3.2. Blood Metabolic Parameters

2,4-TZD treatment significantly decreased NEFA, BHBA, and fatty acids available in plasma and increased glucose in blood (Figure 3; Table 2). Other parameters measured in plasma were not affected by 2,4-TZD treatment (Table 2). 

Results of correlation analysis between metabolic parameters, DMI, and milk-related parameters are available in Appendix A. Milk production was positively associated with DMI, glucose, and cholesterol level in blood, but was negatively associated with the level of urea in blood. Milk fat percentage was negatively associated with glucose and cholesterol in blood but was not associated with the level of NEFA or BHBA.

### 3.3. Adipocytes Size Distribution

Adipocyte area was affected by treatment × time (Figure 4). 2,4-TZD treatment had a significant increase in the median area of adipocytes from Day 13 to Day 26 of treatment, while no changes were detected for the CTR goats. The effect was due to an increased frequency of large adipocytes in TZD vs. CTR goats, whereas the frequency of small-to-medium-sized adipocytes decreased (Figure 4).

### 3.4. Gene Expression

#### 3.4.1. Subcutaneous Adipose Tissue

Relative to CTR, 2,4-TZD treatment decreased the expression of *CD36,* coding for a major protein involved in LCFA import, but increased the expression of *GAPDH* and *IRS1*, tended to increase the transcript abundance of *SLC2A4*, and tended to decrease the expression of *YWHAZ* (Table 3). Among transcripts associated with glyceroneogenesis, we only detected a tendency (treatment × time *p* = 0.08) for a higher expression of *PDK4* in TZD vs. CTR at the end of the trial.

#### 3.4.2. Mammary Epithelial Cells

Table 4 reports the results for the abundance of transcripts in mammary epithelial cells. The isolation of cells using cytokeratin 8 antibody significantly enriched cells expressing the specific epithelial marker *KRT8*; however, contrary to expectations, the expression of mammary-specific genes such as *CSN3*, *LALBA*, and *FABP3* was higher in negative-cytokeratin 8 isolated cells compared with their positive counterparts. 

Almost all transcripts measured were significantly affected by the cell isolation, with *PPARG, GAPDH, NFE2L2, VLDLR,* and *SREBF1* being more abundant in cytokeratin 8-positive vs. cytokeratin 8-negative cells, and *LPL, CD36,* and *SCD1* were more abundant in cytokeratin 8-negative vs. cytokeratin 8-positive cells. None of the measured transcripts were significantly affected by 2,4-TZD treatment except *SCD1*, where 2,4-TZD treatment tended (*p* = 0.06) to decrease its expression in both cells types, and *ACACA*, where expression tended to be increased (*p* = 0.10) by 2,4-TZD treatment in cytokeratin 8-negative cells.

### 3.5. Fatty Acid Profile in Milk

We detected 46 fatty acids in the goat milk (Appendix A). 2,4-TZD treatment significantly (*p* < 0.05) increased the percentage in milk of saturated fatty acids stearic acid (C18:0) and behenic acid (C22:0) and several unsaturated fatty acids, including *cis-7* C14:1, *cis9* C16:1, (unspecified) *trans* C16:1, *trans-9, cis-12, cis-15* C18:3 (Appendix A). Furthermore, *all-trans* linolenic acid was increased by 2,4-TZD treatment, but the same treatment decreased the proportion of *cis-12* C18:1 and C19:1 (Appendix A). A significant or a tendency for treatment × time interaction for the percentage of *cis9* C14:1, *trans-16* C18:1, and C20:4*n6* was detected, whereas a larger proportion over time was detected in TZD vs. CTR (Appendix A). 

When considering the g of FA/milking, only a numerically lower oleic acid (C18:1; *p* = 0.13) and linoleic acid (C18:2; *p* = 0.11) and numerically higher behenic acid (C22:0; *p* = 0.10) in TZD vs. CTR goats were detected (Appendix A). Among all desaturation indexes calculated, the Δ9 18:1 desaturation index was significant lower in TZD vs. CTR (Appendix A).

## 4. Discussion

### 4.1. 2,4-TZD Induces Greater Glucose Utilization in the Adipose Tissue but Did Not Activate PPARγ

The thiazolidinedione family of molecules, including rosiglitazone and pioglitazone, are potent activators of PPARγ, which is a key player in insulin sensitivity, glucose metabolism, lipid homeostasis, and adipogenesis [19,20,21]. In dairy cows, the use of thiazolidinedione molecules during the peripartal period, including 2,4-TZD and pioglitazone, consistently decrease blood NEFA in early lactation [7,22], except when used in non-pregnant dry cows [11]. The decrease of NEFA by thiazolidinedione molecules is a consequence of a positive effect on insulin sensitivity [15]. We did not measure insulin sensitivity in the present experiment, but data indicate a high insulin sensitivity in TZD vs. CTR goats. The improved insulin sensitivity in TZD vs. CTR goats is supported by large decrease of NEFA and, in adipose tissue, an upregulation of *IRS1*, a gene related to insulin signaling [23], a tendency for a higher expression of *SLC2A4*, coding for the insulin-regulated glucose transporter, and a larger transcription of *GAPDH*, a gene positively induced by insulin in rodents [24]. Increased insulin sensitivity by 2,4-TZD treatment was also suggested by data from our prior experiment [9]. The likely increase of insulin sensitivity in goats treated with 2,4-TZD in the present experiment is also supported by an increased size of adipocytes, considering that insulin induces lipogenesis [20].

Different from our prior experiment [9], in the present experiment, blood glucose was increased by 2,4-TZD treatment, corroborating prior observations in dairy cows [7,11,22]. The reason for the increased glucose in blood by 2,4-TZD despite a possible larger insulin sensitivity is unclear; however, the larger concentration of glucose together with gene expression data (*SLC2A4* and *GAPDH*) indicated a larger import and utilization of glucose by the adipose tissue in TZD vs. CTR goats. 

Glucose in adipose tissue of ruminants can be used for fatty acid synthesis, including the provision of NADPH even though acetate is the main source [25]. Larger glucose in plasma after treatment with 4 mg 2,4-TZD/kg BW was also detected in early postpartum dairy cows [6] or in non-pregnant dry dairy cows [11]. The use of pioglitazone prepartum, but not postpartum, also increased glucose levels in dairy cows [7]. These data contrast what is generally observed in humans since thiazolidinedione molecules are classical glucose-lowering drugs. It is possible that the lower plasma NEFA by thiazolidinedione positively affected liver gluconeogenesis [26], increasing glucose in plasma. In the experiment by Smith and collaborators [6], 2,4-TZD tended to lower lactose yield. Similar to Yousefi and collaborators [7], we did not observe any effect of 2,4-TZD on lactose yield in our experiment. The above data indicate that a decrease in utilization of glucose by the mammary gland is not the main cause of the observed increase in hematic glucose.

Association of *GAPDH* expression/activity with adipogenesis has been documented in rodents [27]. The reason for such association remains elusive; however, the reaction of this enzyme is reversible producing dihydroxyacetone phosphate that is then used to produce glyceraldehyde 3-phosphate via *GPD1* through the glyceroneogenesis pathway [28]. Glyceroneogenesis in adipose tissue increases in rodent pre-adipocytes treated with rosiglitazone or pioglitazone via induction of expression of phosphoenolpyruvate carboxykinase [29]. However, 2,4-TZD failed to affect the expression of *PCK1* in our experiment, similar to data in bovine [11]. 

Our original hypothesis was that 2,4-TZD would have activated PPARγ in well-fed animals. Data clearly indicated that 2,4-TZD failed to activate PPARγ in the adipose tissue. The classical PPARγ target gene *LPL* or previously indicated target genes, such as *SREBF1* or *SCD1* [1], were not affected by 2,4-TZD treatment, confirming prior data in cows [30] and goats [9]. Thus, we cannot attribute the decrease in NEFA and the effect on glucose metabolism-related genes in the adipose tissue to PPARγ activation. Further support for inadequate activation of PPARγ by 2,4-TZD is provided by the lack of increase in the frequency of newly formed small adipocytes compared to CTR since PPARγ is a key player in adipogenesis [19]. 

The tendency for transcription of *PDK4* to be increased by 2,4-TZD in the adipose tissue in our experiment is of interest. This enzyme plays a crucial role in glucose metabolism and overall energy homeostasis by inhibiting pyruvate dehydrogenase, providing pyruvate for the formation of oxaloacetate that can be used for glyceroneogenesis [31]. Furthermore, it is a well-established PPARβ/δ target gene in monogastric animals [31] and in dairy cows [32]. Among PPAR isotypes, activation of PPARβ/δ induces a reduction of NEFA in porcine [33]. It is possible that 2,4-TZD activated PPARβ/δ in the adipose tissue in our experiment. However, 2,4-TZD did not affect the expression of *PDK4* in liver (see companion paper [13]).

Overall, our data suggested that 2,4-TZD increased the use of glucose by the adipose tissue, likely for glyceroneogenesis and, maybe, de novo fatty acid synthesis. The latter is partly supported by an increase in the size of adipocytes together with a numerically larger expression of *FASN* and a concomitant decreased expression of *CD36*, important for preformed fatty acid import [34]. However, the effect of 2,4-TZD on the adipose tissue appears to be PPARγ-independent. Although not possible to support with the present data, other mechanisms for the effect of 2,4-TZD on adipose tissue, such as the effect on mitochondria, the activation of AMPK, and/or heat shock response, are possible as previously reviewed [35].

### 4.2. 2,4-TZD Does Not Affect Milk Fat Synthesis

Our original hypothesis was that 2,4-TZD would have increased milk fat synthesis by activating PPARγ. We did not observe any effect of 2,4-TZD on milk fat synthesis. The lack of effect on milk fat synthesis might be indicating a positive effect of 2,4-TZD on milk fat synthesis, considering the large decrease of LCFA available for the mammary gland. Based on our estimates the decrease in NEFA lowered the available LCFA for the mammary gland, including the ones from the triglycerides present in the VLDL, up to 17%. Milk fat is synthesized with about 50% of fatty acids coming from the preformed LCFA present in blood [8]. A positive association between NEFA in plasma and milk fat has been reported in cows [8]. This appeared to be the case from prior studies in periparturient dairy cows where a decrease in milk fat was detected as a consequence of lowered NEFA by 2,4-TZD or pioglitazone treatment [6,7]. NEFA concentration is, however, high during early postpartum (reaching values >1 mM) and the uptake of NEFA by the mammary gland from the blood is highly dependent from its concentration [36]. Despite the above data, it has been determined that more than 98% of stearate in milk fat come from the blood TAG-rich lipoproteins [8]. Thus, a change in NEFA might not have affected milk fat in our experiment due to the relatively low NEFA (< 0.5 mM) as a consequence of the goats being in mid-lactation stage. The lack of any association between NEFA and the percentage of milk fat in our experiment corroborate the above findings. 

We observed only a minor effect of 2,4-TZD on fatty acid composition of the milk, including LCFA with potential positive effects on human health. Among saturated fatty acids affected by 2,4-TZD in our study, stearic acid (C18:0) and behenic acid (C22:0) are known to decrease LDL or total cholesterol in humans and mice [37,38]. Among unsaturated LCFA, proportionally more abundant in TZD vs. CTR goats, punicic acid (C18:3c9,t11,c13) can aid in combating metabolic syndrome [39], and n-7 trans-palmitoleate, abundant in dairy products (possibly the major isomers in our trans-palmitoleate), has been associated with a preventive role in diabetes [40].

The above data indicated that the lowered NEFA did not affect yield of milk fat and only slightly changed milk fat composition. Thus, the data indicated no effect of 2,4-TZD on milk fat synthesis. Gene expression data in MEC support such a conclusion. Despite the fact that the use of the cytokeratin 8 antibody did not enrich milk-secreting MEC from somatic cells, in both cells types isolated, none of the PPARγ target genes were affected by 2,4-TZD, corroborating our prior findings [9]. 2,4-TZD also did not affect the expression of *NFE2L2* and *NRF1,* which code for transcription factors involved in mitochondrial biogenesis and anti-oxidant response [41] and have been previously shown to be upregulated by pioglitazone in bone marrow neuroblasts [42].

Unexpectedly, 2,4-TZD treatment decreased the expression of *SCD1* in MEC. This gene codes for a protein that plays a key role in milk fat synthesis [34]. Inhibition of this protein is associated with a decrease in milk fat synthesis and its expression is consistently decreased during milk fat depression, as previously reviewed [8]. The decrease in expression of *SCD1* in our experiment translated also in a decreased desaturation of stearate to oleic acid. Our data, however, indicated that SCD1 activity is not essential to maintain overall milk fat synthesis in goats, which is contrary to previous reports [8,34] but supports conclusions from a prior study in dairy cows [43].

## 5. Conclusions

Despite having animals in a good body condition and fed adequate levels of vitamin A, we failed to detect any increase in milk fat synthesis or expression of related genes in MEC by 2,4-TZD. The TZD treatment had, however, a large effect on the adipose tissue considering the decreased level of NEFA in blood and increased adipocyte size, while affecting the expression of genes related to insulin signaling and glucose metabolism. Our data clearly indicated that 2,4-TZD has a strong effect on adipose tissue but does not activate PPARγ. This finding undermined the possibility of testing our original hypothesis.

## Figures and Tables

**Figure 1 vetsci-06-00045-f001:**
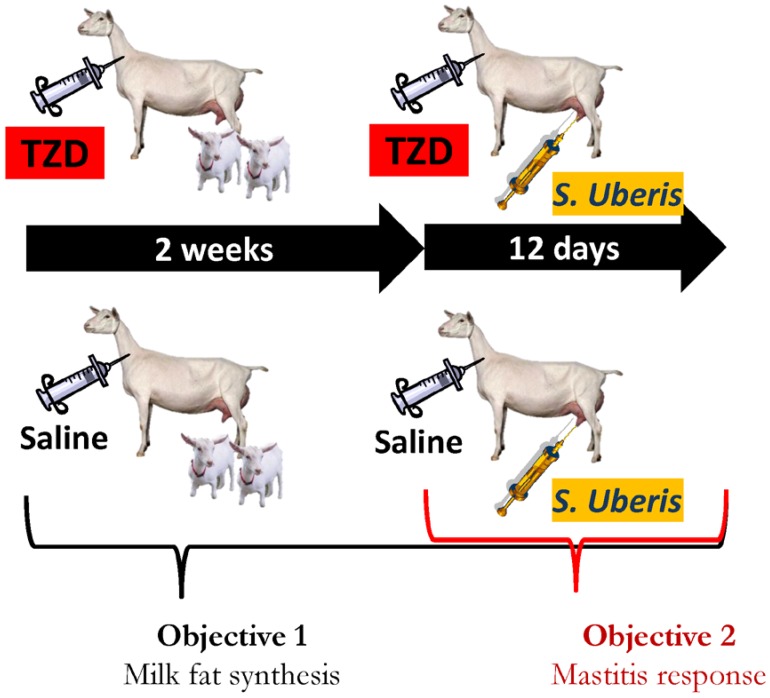
Experimental design. Twelve Saanen dairy goats were randomly assigned to receive a daily injection of 8 mg/kg BW of 2,4-thiazolidinedione (TZD) in 10 mL saline or only saline (*n* = 6/treatment) for 26 days. Kids (2 per goat) were kept nursing the goats for the first 10 days of the experiment. After 2 weeks of treatment the goats received an intramammary infusion of 7 × 10^8^ cfu of *Strep. uberis* in the right half of the mammary gland. All the TZD and three CTR goats were euthanized at the end of the experiment. The whole experiment was used to generate data for Objective 1 (present manuscript) and for Objective 2 in the companion manuscript [13].

**Figure 2 vetsci-06-00045-f002:**
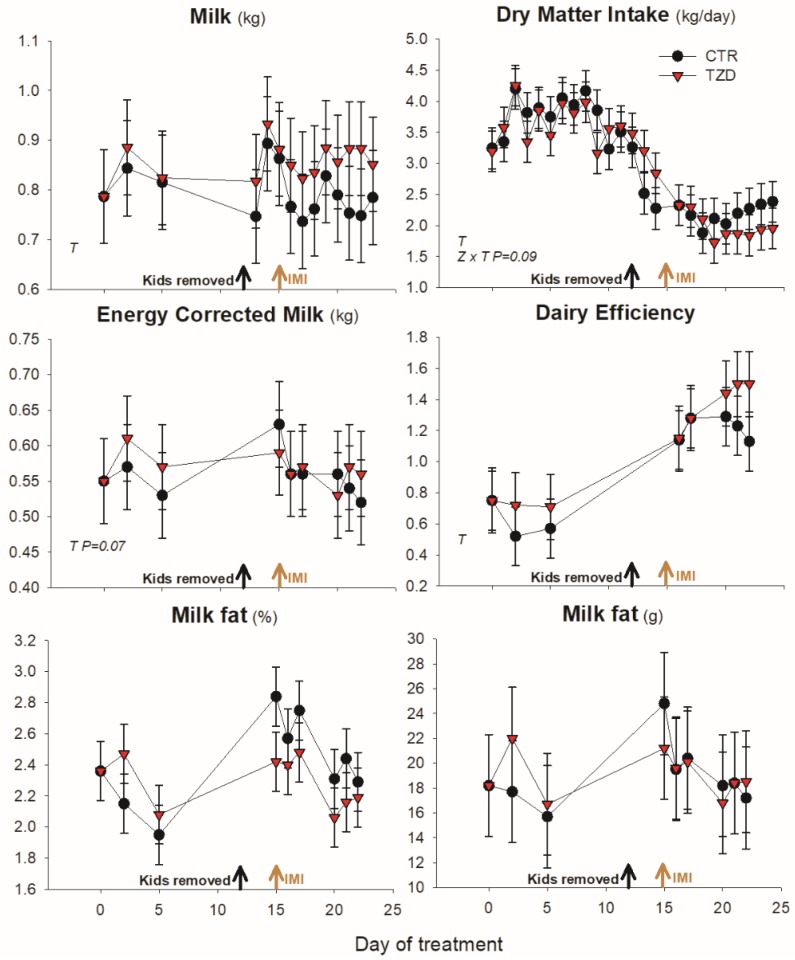
Milk yield (kg produced by the left mammary gland in the AM milking), energy corrected milk, dairy efficiency, daily dry matter intake, milk fat and milk yield (produced by the left mammary gland in the AM milking) in goats receiving daily intra-jugular injection of 2,4-thiazolidinedione (TZD) or saline (CTR). Arrows indicate the time of kid removal and intramammary infusion of *Strep. uberis* in the right half of the mammary gland (IMI). Letters in the graph denote significant (*p* < 0.05) effects of time (T), treatment (Z), and interactions (Z × T). The reported *p*-value is for tendencies. Error bars denote SEM.

**Figure 3 vetsci-06-00045-f003:**
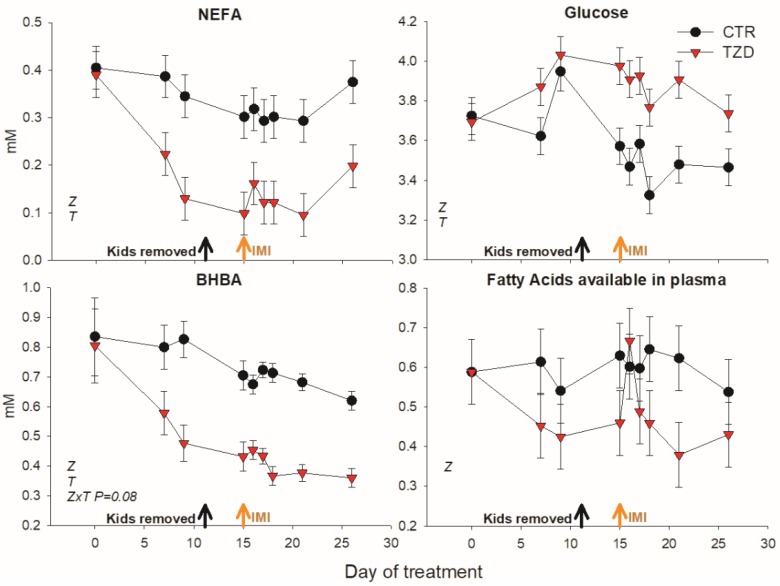
Non-esterified fatty acids (NEFA), glucose, β-hydroxybutyrate (BHBA), and estimated fatty acids present in plasma (mM NEFA + 3× mM TAG) in goats receiving daily intra-jugular injection of 2,4-thiazolidinedione (TZD) or saline (CTR). Arrows indicate removal of kids and time of intramammary infusion of *Strep*. *uberis* in the right half of the mammary gland (IMI). Letters in the graph (a,b) denote significant effects of time (T), treatment (Z), and interactions (Z × T). The reported *p*-value is for tendencies. Error bars denote SEM.

**Figure 4 vetsci-06-00045-f004:**
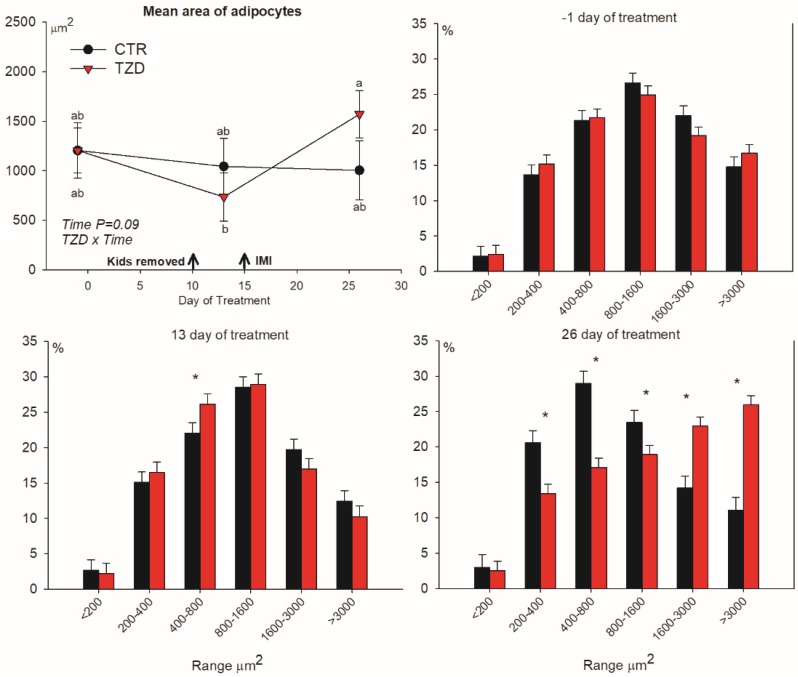
Median area of adipocytes and frequency (%) in each specific range in area of adipocytes in subcutaneous tissue collected from the tail-head of goats receiving daily intra-jugular injection of 2,4-thiazolidinedione (TZD) or saline (CTR). Arrows indicate removal of kids and time of intramammary infusion of *Strep*. *uberis* in the right half of the mammary gland (IMI). Letters in the graph (a, b) denote significant effects of time and interactions (Z × T). Asterisks (*) denote differences (*p* ≤ 0.05) between treatments in the same range of area of adipocytes. Error bars denote SEM.

**Table 1 vetsci-06-00045-t001:** Milk components in lactating goats injected daily with 2,4-thiazolidinedione (TZD) or saline (CTR).

Parameter	Group	Day of Treatment									SEM	*p*-Value ^1^		
		0	2	5	15	16	17	20	21	22		Z	Time	Z × T
Protein %	CTR	2.32	2.29	2.22	2.32	2.35	2.28	2.41	2.41	2.39	0.09	0.30	0.10	0.78
	TZD	2.32	2.26	2.19	2.18	2.18	2.17	2.21	2.21	2.19	0.09			
Protein g ^2^	CTR	18.3	19.4	18.2	20.3	18.2	17.0	19.2	18.4	18.1	2.5	0.98	<0.05	0.76
	TZD	18.3	20.4	17.9	18.9	18.2	17.5	18.4	19.2	19.0	2.5			
Lactose %	CTR	4.45	4.47	4.42	4.38	4.40	4.35	4.34	4.35	4.31	0.07	0.45	<0.05	0.23
	TZD	4.45	4.68	4.51	4.33	4.37	4.41	4.42	4.39	4.37	0.07			
Lactose g ^2^	CTR	34.9	37.2	36.0	37.8	33.7	32.1	34.2	32.7	32.2	4.3	0.61	<0.05	0.61
	TZD	34.9	40.3	36.6	37.3	36.4	35.5	36.9	38.1	37.9	4.3			
SNF ^3^%	CTR	7.66	7.61	7.55	7.53	7.51	7.45	7.48	7.51	7.46	0.31	0.98	0.44	0.63
	TZD	7.66	7.87	7.61	7.42	7.44	7.46	7.46	7.43	7.40	0.31			
SNF g ^2^	CTR	60.2	64.6	61.5	65.4	57.8	55.1	59.2	56.8	56.0	7.6	0.69	<0.05	0.67
	TZD	60.2	69.3	62.0	64.1	62.1	60.2	62.1	64.5	64.2	7.6			

^1^ Z = 2,4-thiazolidinedione effect; T = time; Z × T = 2,4-thiazolidinedione × time effect. ^2^ Grams produced by the left mammary gland in the AM milking. ^3^ Solid non-fat.

**Table 2 vetsci-06-00045-t002:** Plasma concentration of metabolic parameters in dairy goats injected daily with 2,4-thiazolidinedione (TZD) or saline (CTR).

Parameter	Group	Day of Treatment									SEM	*p*-Value ^1^		
		0	7	9	15	16	17	18	21	26		Z	Time	Z × T
Cholesterol, mM	CTR	2.55	2.30	2.27	2.18	2.28	2.31	2.33	2.40	2.32	0.24	0.38	0.18	0.70
	TZD	2.55	2.73	2.56	2.40	2.51	2.61	2.61	2.62	2.61	0.24			
∆FA, % ^3^	CTR	0.0	5.1	−6.8	5.0	3.8	0.8	9.1	7.7	−6.6	8.5	0.06	0.88	0.90
	TZD	0.0	−13.2	−14.5	−9.4	−4.9	−8.1	−9.7	−17.2	−12.0	8.5			
TAG, mM	CTR	0.14	0.15	0.16	0.19	0.17	0.18	0.19	0.19	0.13	0.02	0.89	0.30	0.82
	TZD	0.14	0.14	0.15	0.19	0.19	0.19	0.18	0.16	0.15	0.02			
Creatinine, µM	CTR	71.9	77.3	77.1	71.9	72.6	70.2	70.4	72.2	75.8	2.7	0.98	<0.05	0.94
	TZD	71.9	76.6	77.6	70.7	72.2	71.7	70.6	72.4	76.1	2.7			
Urea, mM	CTR	6.73	6.32	4.59	5.97	6.81	5.61	6.54	6.45	8.05	0.45	0.25	<0.05	0.19
	TZD	6.73	6.40	5.16	5.29	6.08	5.08	5.61	5.71	6.22	0.45			

^1^ Z= 2,4-thiazolidinedione effect; T = time; Z × T = 2,4-thiazolidinedione × time effect. ^3^ Percentage variation of total available fatty acids (as mM of NEFA + (3× mM of TAG)) compared to time 0.

**Table 3 vetsci-06-00045-t003:** mRNA abundance of selected genes in adipose tissue of goats treated with 2,4-thiazolidinedione (TZD) or saline (CTR). Samples were *n* ≥ 5 for both groups, except for the day 26 (*n* = 3 for CTR; *n* = 6 for TZD).

Gene	Group	Time			SEM ^1^	*p*-Value ^2^		
		−1	13	26		Z	Time	Z × T
Fatty Acids Import and De Novo Fatty Acid Synthesis								
*CD36*	CTR	7169.8	6987.6	6233.3	681.3	0.03	0.03	0.03
	TZD	7169.8	3362.5	5781.9				
*LPL*	CTR	122.2	130.5	244.6	52.1	0.62	0.04	0.90
	TZD	122.2	118.0	206.7				
*SLC27A6*	CTR	0.09	0.12	0.11	0.03	0.84	0.15	0.94
	TZD	0.09	0.14	0.11				
*VLDLR*	CTR	2.3	2.0	1.0	0.9	0.67	0.48	0.25
	TZD	2.3	1.1	2.7				
*ACACA*	CTR	37.6	26.6	48.5	24.3	0.33	0.51	0.14
	TZD	37.6	88.6	45.6				
*FASN*	CTR	473.3	47.7	58.6	439.9	0.13	0.29	0.18
	TZD	473.3	163.1	1328.2				
*SCD1*	CTR	275.2	500.8	410.9	333.4	0.67	0.16	0.54
	TZD	275.2	917.0	329.3				
Insulin Signal and Glucose Metabolism								
*GAPDH*	CTR	159.2	125.6	91.2	41.7	0.01	0.25	0.08
	TZD	159.2	271.2	195.9				
*INSR*	CTR	7.8	5.2	3.7	1.9	0.12	0.37	0.21
	TZD	7.8	5.8	9.4				
*IRS1*	CTR	9.2	6.8	4.6	2.0	0.04	0.42	0.05
	TZD	9.2	7.4	12.5				
*SLC2A4*	CTR	4.5	5.6	5.1	1.6	0.09	0.10	0.40
	TZD	4.5	8.9	7.3				
*YWHAZ*	CTR	167.5	191.8	149.0	23.9	0.07	0.50	0.04
	TZD	167.5	101.4	150.1				
Glyceroneogenesis								
*PCK1*	CTR	8.0	9.8	15.0	4.2	0.67	0.01	0.47
	TZD	8.0	8.7	21.0				
*PDK4*	CTR	1.8	1.2	1.1	0.4	0.66	0.03	0.08
	TZD	1.8	0.8	2.1				
*GPD1*	CTR	71.5	109.1	90.8	33.9	0.45	0.07	0.71
	TZD	71.5	145.8	116.4				
Transcription Regulation								
*PPARG*	CTR	43.5	41.7	49.5	9.6	0.81	0.19	0.70
	TZD	43.5	37.3	58.3				
*SREBF1*	CTR	16.1	19.3	23.0	16.7	0.61	0.38	0.61
	TZD	16.1	14.8	44.7				

^1^ The highest SEM is shown. ^2^ Z = 2,4-thiazolidinedione effect; T = time; Z × T = 2,4-thiazolidinedione × time effect.

**Table 4 vetsci-06-00045-t004:** Transcript abundance of selected genes normalized by the geometric mean of *RPS9*, *UXT*, and *MRPL39* in mammary epithelial cells (MEC) in goats receiving 2,4-thiazolidinedione (TZD) or saline (CTR).

Gene	Group	Cell ^1^		SEM	*p*-Value ^2^		
		Neg	Pos		Z	Cell	Z × Cell
Mammary Epithelial-Specific Genes							
*CSN3*	CTR	6324	3877	875	0.46	<0.05	0.42
	TZD	5050	3326	911			
*FABP3*	CTR	20.0	8.5	3.0	0.57	<0.05	0.41
	TZD	16.1	7.8	3.2			
*LALBA*	CTR	1341	578	233.4	0.56	<0.05	0.56
	TZD	1065	501	251.7			
*KRT8* (Ln) ^3^	CTR	1.78	2.50	0.27	0.30	<0.05	0.64
	TZD	1.48	2.03	0.29			
Fatty Acid Transport and Synthesis							
*LPL*	CTR	9.68	4.87	1.52	0.70	<0.05	0.77
	TZD	10.8	5.19	1.68			
*CD36*	CTR	1300	979	104	0.30	<0.05	0.87
	TZD	1445	1094	115			
*VLDLR* (Ln)	CTR	0.57	0.84	0.13	0.21	<0.05	0.31
	TZD	0.43	0.53	0.14			
*ACACA*	CTR	0.29	0.58	0.09	0.99	0.09	0.10
	TZD	0.43	0.43	0.10			
*SCD1*	CTR	5.84	5.03	0.68	0.06	0.02	0.53
	TZD	4.19	2.89	0.71			
Transcriptional Regulation							
*NFE2L2*	CTR	51.2	62.0	12.3	0.30	<0.05	0.50
	TZD	66.4	83.7	12.6			
*NRF1*	CTR	0.15	0.17	0.03	0.69	0.50	0.91
	TZD	0.17	0.18	0.03			
*PPARG* (Ln)	CTR	-2.73	−1.94	0.33	0.38	<0.05	0.50
	TZD	-2.20	−1.71	0.35			
*SREBF1*	CTR	0.85	1.17	0.13	0.45	<0.05	0.79
	TZD	0.74	1.01	0.14			
Other							
*GAPDH* (Ln)	CTR	3.63	4.07	0.31	0.39	<0.05	0.47
	TZD	3.85	4.56	0.33			
*IL8* (Ln)	CTR	4.8	4.9	0.6	0.31	0.17	0.40
	TZD	5.5	5.9	0.6			
*YWHAZ*	CTR	31.6	40.7	8.3	0.64	0.19	0.61
	TZD	39.1	43.3	8.3			

^1^ Pos = cell positively isolated using cytokeratin 8; Neg = the cells that remained after the isolation of cytokeratin + cells. ^2^ Z = 2,4-thiazolidinedione effect, T = time, Z × T = 2,4-thiazolidinedione × time effect. ^3^ Data were transformed prior to statistical analysis and showed as Ln (natural logarithm).

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
