# Peer review of "2,4-Thiazolidinedione in Well-Fed Lactating Dairy Goats: I. Effect on Adiposity and Milk Fat Synthesis"

_vetsci, 2019, doi:10.3390/vetsci6020045_

Round 1
Reviewer 1 Report
The manuscript is interesting and well presented. Only some minor consideration.
Line 111. I did not understand the rational of the injection of sterile physiological saline. Also for animals is it considered a placebo effect?
line 112: explain MW
line 113 mm/l? mm/dl?
Author Response
Comments and Suggestions for Authors
The manuscript is interesting and well presented. Only some minor consideration.
Line 111. I did not understand the rational of the injection of sterile physiological saline. Also for animals is it considered a placebo effect?
AU: The injection of the saline is to account for the effect of the saline itself and to account for the effect of injection.
line 112: explain MW
AU: Addressed by spelling our the acronym
line 113 mm/l? mm/dl?
AU: mM is equivalent to mmol/L
Reviewer 2 Report
General comments:
The manuscript by Jaaf et al. was well written. Though the idea is not novel and the hypothesis in the current study was based on a previous study from the same group. The authors hypothesized that continuous activation of PPARγ by 2,4-TZD increases milk fat synthesis in dairy goats in good body condition and supplemented with an adequate amounts of vitamin A. However, they failed to detect any increase in milk fat synthesis or expression of related genes in mammary epithelial cells (MEC) by 2,4-TZD. Though many non-significant results were reported in the manuscript, the experimental design is sound and the authors did huge amount of work including blood/milk sample assays and RT-qPCR of adipose tissue and MEC. Statistical analyses are clearly described, results are adequate, and discussion is thorough. Though lots of negative findings, the current study showed that 2,4-TZD has a strong anti-lipolysis effect on adipose tissue. A few minor points need to be addressed.
Specific comments:
Line (L) 86: Figure 1. What is the rationale of treatment 8mg/kg BW of 2,4-TZD? The preliminary data in determining the concentration should be provided in supplemental material. Or you could cite a reference. The objective 2 should be removed from this figure since it has been published in a companion manuscript. No need to talk too much about objective 2 throughout the text.
L134-139: Did you do duplicates in all the assays? Intra-CV and inter-CV should be mentioned here.
L264: Discuss more about the potential effects of fatty acids with different length of carbon chains and unsaturation levels.
Author Response
AU: we thanks the reviewer for the comments
Line (L) 86: Figure 1. What is the rationale of treatment 8mg/kg BW of 2,4-TZD? The preliminary data in determining the concentration should be provided in supplemental material. Or you could cite a reference. The objective 2 should be removed from this figure since it has been published in a companion manuscript. No need to talk too much about objective 2 throughout the text.
AU: Added the reference for the dose. Even though we understand the objection of the reviewer, we think the reader needs to have an overview of the whole experiment that comprises 2 main objectives. We think that the overview of the experimental design including both objectives should be provided in the first companion manuscript. The other manuscript (i.e., the companion manuscript) follow the present manuscript. Thus, we prefer to leave the Figure in the present manuscript. We also think that we need to clearly indicate to the reader that the goats received an intramammary infection because some of the data (e.g., milk and milk components, pattern of adipocyte size) includes also data point after the intramammary infection and the figures report the moment of intramammary infection.
L134-139: Did you do duplicates in all the assays? Intra-CV and inter-CV should be mentioned here.
AU: The Intra-CV and inter-CV are reported in the companion manuscript. We have added the reference to the companion manuscript indicating that the above can be found in the supplementary file of the companion manuscript. The assays were not run in duplicate.
L264: Discuss more about the potential effects of fatty acids with different length of carbon chains and unsaturation levels.
AU: it is unclear the request from the reviewer. However, we have discussed the possible effect of our treatment on fatty acid composition of milk fat and the potential effects on human health. The origin of those unsaturated fatty acids is somewhat unclear and our limited data do not allow to provide substantiated inferences on possible bio-hydrogenation and unsaturation that happened before the fatty acids entered the mammary gland. The decreased in delta-9 desaturation might indicate that the production of punic acid and is outside the mammary. Also, it is unclear where the formation of trans-palmitoleic happened; however, we have added a refernece indicating that might be deriving from partial beta-oxidation of vaccenic acid (however, this is quite speculative). We have also discussed the possible reason for the lower desaturation index observed when goats were treated with 2,4-TZD.